# Validation of the LittlEARS® Questionnaire in Hearing Maltese-Speaking Children

**Pauline Miggiani [1], Frans Coninx [2] and Karolin Schaefer [3],***

1  Audiology Clinic, Mater Dei Hopsital, MSD 2090 Msida, Malta; paulinemig@gmail.com
2  IfAP, Institut für Audiopädagogik, 42697 Solingen, Germany; f.coninx@ifap.info
3  Department of Special Education and Rehabilitation, Faculty of Human Sciences, University of Cologne, 50931 Köln, Germany
*  Correspondence: karolin.schaefer@uni-koeln.de

**Abstract:** Objectives: To adapt the LittlEARS® Auditory Questionnaire into the Maltese language and evaluate the psychometric properties of the Maltese version of the questionnaire for hearing children. Methods: The English version of LittlEARS® Auditory Questionnaire was adapted into Maltese using a translation/back translation procedure. In this cross-sectional study, a total of 398 parents of normal hearing children aged between 5 days and 36 months completed the Maltese version of LittlEARS®. Psychometric validation was performed through scale analysis, item analysis, and analysis of reliability and validity. A non-linear regression model was derived to obtain normative data for expected and minimum values of total scores from the questionnaire according to age. Results: Predictive accuracy (Guttman's lambda) was 0.921, the Cronbach's alpha coefficient value was 0.921, and the split-half reliability coefficient was 0.949. The Pearson correlation coefficient between scores and age was 0.903. The regression analysis showed that 82% of the variance in the total scores can be explained by age. Norm curves were comparable to the original German data. Conclusion: This study confirmed that the Maltese version of LittlEARS® is a valid and reliable tool to evaluate auditory development in children less than two years of age.

**Keywords:** LittlEARS; auditory questionnaire; normative assessment; hearing screening; Maltese

## 1. Introduction

Over the past 20 years, Universal Newborn Hearing Screening (UNHS) programmes have enabled the identification of and early intervention for newborns with permanent hearing loss [1]. However, this success does not diminish the need for screening older children. It is widely known that not all cases of hearing loss in early childhood are detected in the newborn period. This is mainly due to progressive or late-onset hearing loss [2]. Another screening follow-up before school age is thus recommended by researchers and medical professionals alike [3–5]. This has, in turn, necessitated more and better ways of assessing young children's auditory development.

Structured parental questionnaires have proven to be useful tools in the evaluation of young children. Early, preverbal auditory behaviours cannot always be observed in a clinical setting. Some children are uncooperative in unfamiliar surroundings, whilst others are too young to participate in formal tests [6–8]. Through questionnaires, parents and caregivers can describe quickly and concisely the child's auditory behaviours and responses in various real-life situations. While efficient and cost effective, these tools are primarily available in English. Maltese is the national language of the Maltese islands. It is spoken by over 90% of the population aged 10 years and older [9]. Aside from Maltese, English is the only other official language of the country, with over 62% having a good command of it [9]. The majority of the Maltese population can therefore be classified as bilingual [10]. In general, Maltese is used at home and within the community, whilst English is used in higher educational contexts [11].

The LittlEARS® Auditory Questionnaire (LEAQ) is a parent-based tool that has been translated into more than 15 languages worldwide. It has been recognised internationally as a valid and reliable language-independent measure to assess the early auditory behaviours of hearing infants and toddlers [12]. LEAQ consists of 35 yes/no questions about the child's responses to sounds, particularly linguistic stimuli and early speech development. The items are arranged developmentally, with the earliest-developing behaviours at the beginning of the questionnaire.

The present cross-sectional study aims to translate and validate the LEAQ for hearing Maltese infants and to define Maltese critical score values (expected and minimum). It also aims to provide a tool that may be used as a second hearing screening following the newborn hearing screening.

## 2. Materials and Methods

### 2.1. Participants

In this cross-sectional study, 398 Maltese children aged between 5 days and 36 months were recruited via email from the local general hospital and from 12 childcare centres in Malta. A minimum sample of 373 participants selected randomly from a population of approximately 12,900 (4300 children are born in Malta per year) guarantees a maximum margin of error of 5%, assuming a 95% confidence level. A total of 398 participants were included in the sample. Inclusion criteria included the absence of known disabilities (hearing loss, neurological disorders, and premature babies). The questionnaire was self-administered by the parents and caregivers, and the researcher did not influence the process in any way. To control for typical hearing ranges, all children were first screened using OAEs before parents filled in the questionnaire. Children who did not pass were referred to the relevant audiological services and were omitted from the study. All participants were volunteers and received no compensation for their participation. The study was approved by the local university research ethics committee, UREC (Ref. No. UREC-DP1801016EXT). Data collection was done by Dr Pauline Miggiani between June 2019 and August 2020.

In total, 268 children (0–24 months) were included in the main analysis, of which 131 were male and 137 were female. A total of 130 children (24–36 months) were included to confirm ceiling effects in the third year of life (24–36 months). The age distribution of the children is shown in Table 1.

**Table 1.** Age distribution of participants by gender.

|  | Age (Months) | Males | Females | Total |
|---|---|---|---|---|
|  | 0–2 | 18 | 16 | 34 |
|  | 3–5 | 16 | 18 | 34 |
| Main | 6–8 | 10 | 20 | 30 |
| Analysis | 9–11 | 11 | 20 | 31 |
| (268) | 12–14 | 17 | 18 | 35 |
|  | 15–17 | 16 | 13 | 29 |
|  | 18–20 | 24 | 13 | 37 |
|  | 21–23 | 19 | 19 | 38 |
|  | 24–26 | 22 | 22 | 44 |
| (130) | 27–29 | 11 | 16 | 27 |
|  | 30–32 | 22 | 13 | 35 |
|  | 33–35 | 16 | 8 | 24 |
|  | Total | 202 | 196 | 398 |

### 2.2. Translation

The LittlEARS® Auditory Questionnaire was originally developed in German [12,13]. This version was translated into English, which has subsequently served as the basis for adaptation into several other languages. The English version was adapted into Maltese using the translation/back-translation procedure, recommended by the International Test

Commission [14]. The purpose of the back-translation design was to keep the variable meanings of the test items in the questionnaire and, in addition, to get a linguistically correct version [15].

Two professional translators were recruited to translate the test items from English into Maltese. The adaptation into the Maltese language was carried out in two phases. The first was the translation phase, whilst the second was the evaluation phase. This was done by means of an expert appraisal method, which ensures that the translated version of the text is linguistically equivalent and is of the best professional quality [16]. The translation/back-translation included (1) direct translation from English (source language) into Maltese (target language), (2) back-translation from the target language (Maltese language version) into English, and (3) comparing the original English and Maltese back-translations [16].

The main adaptation that was needed was the development of a separate questionnaire for male and female respondents to minimise confusion, as the gender-neutral sentences were too long and complex to follow. For instance, Item 2 in LittlEARS, 'Does your child hear somebody speak?' would be translated as 'Ibnek/bintek jisma'/tisma' lil xi ħadd jitkellem?'. Hence, two versions were necessary to simplify the text for the respondents.

An evaluation of the translations was then carried out. The expert appraisal method provides evidence regarding the quality of the translated version and recommends ways to improve the final version [16]. The appraisers were each provided with a set of evaluation forms to ensure that the evaluation was systematic and orderly. The task of the appraisers was to compare both English and Maltese versions of each test item, including the examples, to assess the extent to which both versions measured the same auditory behaviour.

Following the review, 4 statements were identified for revision. In the example of Item 1, 'Smiles; looks towards source?', 'tal-ħoss' (of the sound) was added to the description to decrease ambiguity. Similarly, in the example of Item 10, 'Musical box by bed; lullaby; water running into the tub', the English word 'lullaby' was also added since the Maltese translation is rarely used. The descriptions of Items 31 and 35 also included the English words 'nursery rhymes' and 'lullaby', respectively for the same reasons.

Parents were asked to fill in the questionnaire by ticking 'yes' if the behaviour was observed at least once and 'no' if the behaviour was never observed. Most of the questions include an example of the behaviour being assessed in the question, thereby increasing the objectiveness of the tool; for example, Question 16 asks, 'Does your child respond to music with rhythmical movements? (The child moves arms/legs to the music).' The total score is the number of 'yes' responses.

The total score is then compared to the expected value and minimal value. The former is the average score of a hearing peer, whilst the latter is the minimum score a hearing child at that age should attain on the LEAQ. If a child scores above the minimum value, there is a high probability (95%) that their auditory development is age appropriate. However, if the minimum score is lower, the child should be assessed further [13].

*2.3. Analysis*

The psychometric properties of the Maltese version were evaluated to confirm the reliability and validity of the translation. The predictive accuracy, the split-half reliability coefficient, the Cronbach's alpha coefficient and the correlation between age and total score were computed for the scale analysis. Correlation between age and item score, index of difficulty, and the discrimination coefficient were computed for the item analysis. The Kolmogorov–Smirnov and the Shapiro–Wilk Tests indicated that the total score distribution does not satisfy the normality assumption ($p \geq 0.05$). For this reason, non-parametric tests were used to analyse the data. A norm curve was generated through quadratic regression analyses.

## 3. Results

### 3.1. Scale Analysis

The predictive accuracy of the Maltese version of the LEAQ was calculated using Guttman's lambda 2. A value of 0.921 was reported, confirming a significant predictability. The split-half reliability coefficient for the Maltese version of the LEAQ was 0.949, indicating that the questionnaire has a high measuring accuracy and reliability. The Cronbach's alpha coefficient value was 0.921 for the Maltese LEAQ. This suggests that the responses from subjects are highly consistent across the questionnaire items. The values of the Cronbach's alpha coefficient seen demonstrate that the items in the Maltese version of LEAQ can reliably differentiate the degree of auditory development in the Maltese children evaluated in this study.

The correlation between age and total score was calculated to obtain information about the ability of the questionnaire to measure age-dependent auditory behaviour. The Pearson correlation coefficient is 0.903, which indicates a high positive correlation between scores and age. As children grow older, their expected scores are higher. This provides evidence for the validity of the Maltese version of the LEAQ.

The psychometric properties of the Maltese version were compared with those of the original German version. A high similarity between the two versions was observed. Table 2 also includes a scale analysis of other languages for comparison.

**Table 2.** Parameter comparisons of the scale analysis and regression equations of LEAQ norm curves across language versions.

| LEAQ Version | Correlation Age + Total Score | Guttman's Lambda | Split-Half Reliability | Cronbach's Alpha | Regression Equation |
|---|---|---|---|---|---|
| Maltese | 0.90 | 0.92 | 0.95 | 0.92 | $y = -0.03x^2 + 2.02x + 5.07$ |
| German | 0.91 | 0.93 | 0.88 | 0.96 | $y = -0.038x^2 + 2.22x + 2.07$ |
| Mandarin | 0.84 | 0.88 | 0.91 | 0.95 | $y = -0.038x^2 + 2.23x + 1.21$ |
| Polish | 0.90 | | | 0.95 | $y = -0.028x^2 + 1.98x - 4.85$ |
| Yoruba | 0.78 | 0.58 | 0.70 | 0.91 | $y = -0.081x^2 + 3.303x + 0.648$ |
| Persian | 0.81 | 0.96 | 0.73 | 0.96 | |
| Turkish | 0.84 | 0.91 | | 0.94 | |
| Multilingual * | 0.89 | 0.92 | 0.94 | 0.96 | $y = -0.038x^2 + 2.163x + 3.470$ |

* [17].

### 3.2. Item Analysis

(a) Item Difficulty

The index of difficulty for each item on the questionnaire is given in Table 3 (Columns 4–5). It displays the ratio of the number of subjects who give a 'yes' response to the whole number of subjects (N = 268). In this study, the indices ranged from 0.30 to 0.99, whilst the original German version ranged from 0.25 to 0.98. This shows that the items of the questionnaire are presented in order of difficulty, from the easiest items indicating basic auditory skills to the most difficult ones demonstrating advanced auditory skills. Questions with a high index of difficulty were kept in the questionnaire to ensure that no child gets a score of zero.

(b) Discrimination Coefficient

The correlation between an individual item and the total score on the questionnaire is referred to as the discrimination coefficient and is shown in Table 3 (Columns 6–7). A high correlation value indicates that the item has a significant impact on the total score. This also helps in differentiating between good and poor performers. For instance, Items 1 and 2 have the lowest correlation coefficients, suggesting that these two items have limited contribution in distinguishing between good and poor performers. On the other hand, the other items show high discrimination values, confirming that the Maltese version of the LEAQ can differentiate between children displaying age-dependent auditory responses.

(c)    Correlation between age and item score

The correlation between age and item score is shown in Columns 2 and 3 in Table 3. This was calculated to check the items' suitability for measuring the age dependency of behaviours. The correlation coefficients range from 0.14 to 0.81. The average correlation is generally moderate. About one third of the items show a strong positive correlation with age ($r \geq 0.7$), while only a few are weakly correlated ($r \geq 0.3$). Items with a low coefficient have limited meaning for the child's age-dependent auditory response. The first four items are intended for measuring auditory behaviours that even very young children can exhibit. These items ensure that no child gets a score of zero. This was observed across several LEAQ versions [7,17–20], and hence, the first four questions were included in the Maltese version.

**Table 3.** Parameter comparison of the item analysis between the Mandarin version and the German version of the LEAQ, reported by Coninx et al. (2009) [17].

| Item No. | Corr. Age + Item Score | | Index of Difficulty | | Discrimination Coefficient | |
|---|---|---|---|---|---|---|
| | **Maltese** | **German** | **Maltese** | **German** | **Maltese** | **German** |
| 1 | 0.19 | 0.21 | 0.99 | 0.98 | 0.04 | 0.25 |
| 2 | 0.14 | 0.10 | 0.97 | 0.98 | 0.08 | 0.16 |
| 3 | 0.32 | 0.30 | 0.93 | 0.94 | 0.60 | 0.37 |
| 4 | 0.31 | 0.26 | 0.96 | 0.93 | 0.48 | 0.37 |
| 5 | 0.46 | 0.41 | 0.86 | 0.95 | 0.43 | 0.51 |
| 6 | 0.31 | 0.47 | 0.93 | 0.84 | 0.57 | 0.59 |
| 7 | 0.53 | 0.44 | 0.82 | 0.83 | 0.73 | 0.54 |
| 8 | 0.37 | 0.13 | 0.76 | 0.82 | 0.72 | 0.24 |
| 9 | 0.50 | 0.52 | 0.75 | 0.81 | 0.75 | 0.66 |
| 10 | 0.47 | 0.43 | 0.88 | 0.80 | 0.60 | 0.55 |
| 11 | 0.37 | 0.47 | 0.93 | 0.78 | 0.63 | 0.58 |
| 12 | 0.63 | 0.69 | 0.82 | 0.74 | 0.79 | 0.76 |
| 13 | 0.57 | 0.59 | 0.81 | 0.74 | 0.82 | 0.73 |
| 14 | 0.15 | 0.33 | 0.89 | 0.72 | 0.61 | 0.29 |
| 15 | 0.63 | 0.67 | 0.71 | 0.71 | 0.90 | 0.76 |
| 16 | 0.62 | 0.66 | 0.72 | 0.70 | 0.88 | 0.75 |
| 17 | 0.71 | 0.64 | 0.60 | 0.69 | 0.82 | 0.76 |
| 18 | 0.69 | 0.76 | 0.60 | 0.64 | 0.84 | 0.81 |
| 19 | 0.69 | 0.63 | 0.62 | 0.63 | 0.93 | 0.71 |
| 20 | 0.73 | 0.80 | 0.62 | 0.59 | 0.92 | 0.86 |
| 21 | 0.63 | 0.50 | 0.60 | 0.55 | 0.90 | 0.75 |
| 22 | 0.80 | 0.81 | 0.53 | 0.52 | 0.79 | 0.87 |
| 23 | 0.75 | 0.80 | 0.49 | 0.51 | 0.80 | 0.85 |
| 24 | 0.81 | 0.81 | 0.47 | 0.50 | 0.52 | 0.87 |
| 25 | 0.75 | 0.73 | 0.45 | 0.42 | 0.70 | 0.78 |
| 26 | 0.74 | 0.79 | 0.37 | 0.42 | 0.67 | 0.81 |
| 27 | 0.71 | 0.75 | 0.41 | 0.40 | 0.52 | 0.79 |
| 28 | 0.70 | 0.73 | 0.37 | 0.40 | 0.66 | 0.77 |
| 29 | 0.73 | 0.64 | 0.46 | 0.39 | 0.75 | 0.70 |
| 30 | 0.70 | 0.77 | 0.34 | 0.39 | 0.60 | 0.80 |
| 31 | 0.69 | 0.70 | 0.40 | 0.38 | 0.58 | 0.72 |
| 32 | 0.70 | 0.70 | 0.40 | 0.34 | 0.52 | 0.72 |
| 33 | 0.39 | 0.63 | 0.76 | 0.32 | 0.37 | 0.62 |
| 34 | 0.69 | 0.71 | 0.30 | 0.27 | 0.40 | 0.65 |
| 35 | 0.69 | 0.62 | 0.37 | 0.25 | 0.40 | 0.57 |
| Average | 0.57 | 0.58 | 0.65 | 0.63 | 0.64 | 0.64 |

The Kruskal–Wallis Test was used to compare mean total scores clustered by age. As seen in Figure 1, the mean total score increases significantly between 1 and 24 months. However, the score remains fairly level after the age of 24 months ($p > 0.05$).

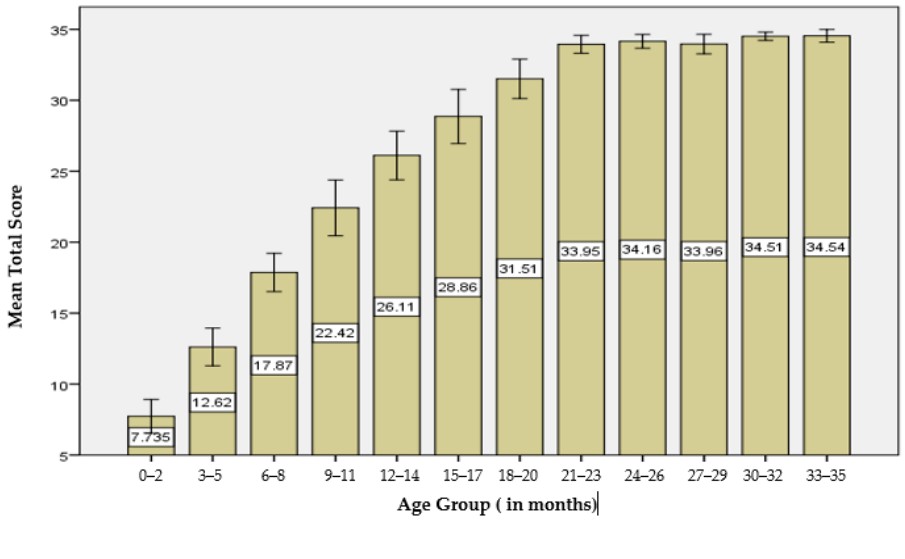

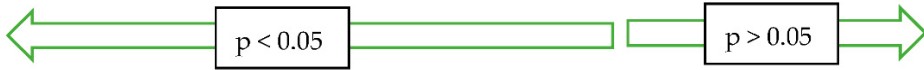

**Figure 1.** Mean total score distribution of hearing Maltese children (N = 268) by age group.

Mean Total Scores by Gender

The mean total score of males does not exceed the mean total score of females ($p = 0.323$) by a statistically significant amount.

*3.3. Generation of a Norm Curve*

To establish a norm curve for the development of auditory behaviours of Maltese children between 0 and 24 months of age, a regression analysis with age as the independent variable and total score as the dependent variable was carried out. A scatter plot of the raw data and the generated quadratic norm curve generated is shown in Figure 2. The minimum and maximum values are also provided. The regression equation for the Maltese sample is $y = -0.03x^2 + 2.02x + 5.07$, where the total score is represented by the variable y, and age is represented by the variable x. The coefficient determination for this model shows that 82% of the variance in the total scores can be explained by age ($R^2 = 0.82$). The Maltese norm curve was also compared with the German norm curve, which was plotted using the regression equation of the German data, $y = -0.038x^2 + 2.22x + 2.07$. As seen in Figure 2, the coefficient of x and $x^2$ are very similar, which explains why the two curves are almost parallel. However, the constant terms (5.07 and 2.07) differ by 3, implying that the Maltese children are scoring 3 points higher, on average, than the children in the German data.

Figure 2 also shows the overall curve of the 15 languages from Coninx et al. [17], derived from the equation, $-0.038x^2 + 2.163x + 3.470$. Similarly, one can note that the coefficients are very similar, and the constants differ very slightly (5.07 and 3.47), confirming how close the Maltese norm curve is to the other 15 languages, validated by Coninx et al. [17].

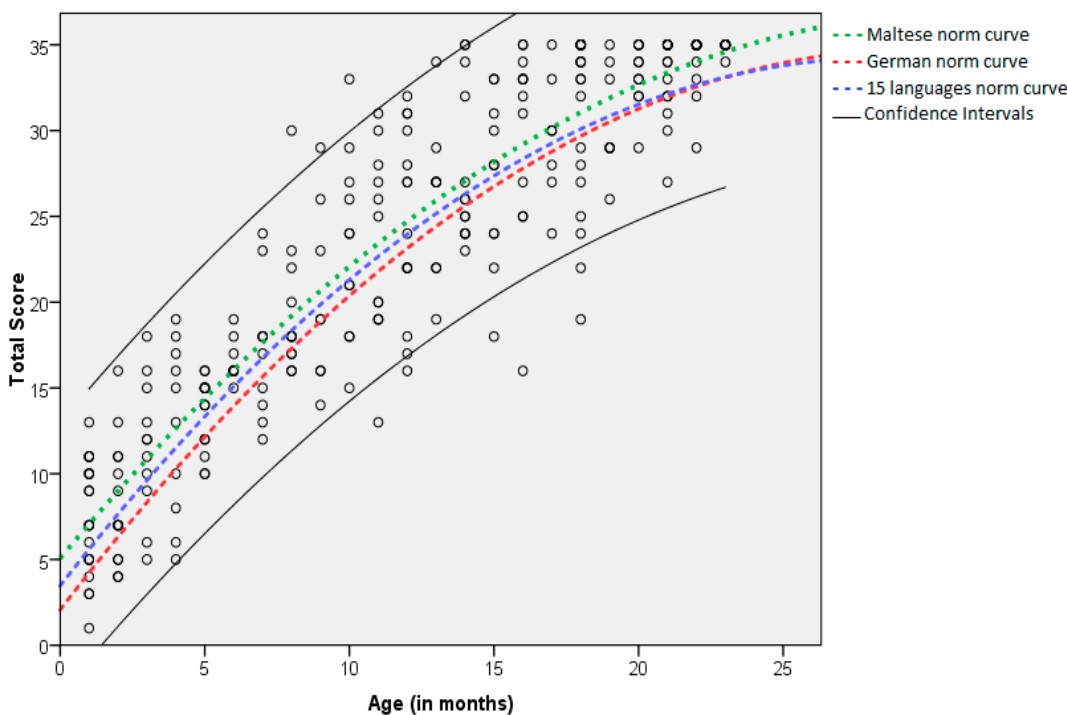

**Figure 2.** Regression curves (quadratic) with age as the independent variable and total scores as dependent variables in comparison to other languages. Note: The green line shows the Maltese normative curve, and the red line shows the original German normative curve. The blue line corresponds to the overall normative curve for 15 languages. The upper and lower black lines display the upper and lower confidence intervals of the Maltese sample. The circles represent the raw data.

## 4. Discussion

### 4.1. Psychometric Properties of the Maltese LEAQ

The aim of this study was to adapt the LEAQ for use with Maltese-speaking parents, caregivers, and professionals. Two measures were included: the psychometric properties of the Maltese version of the LEAQ and the generation of normative data for Maltese children involved in the questionnaire.

Translations of established questionnaires have often been the choice in populations whose first language is not English. The challenge in such a task lies in adapting a tool in a culturally relevant manner whilst maintaining the meaning and intent of the original items. In this study, a back-translation method was used to develop the Maltese version. This is similar to the Polish [16], Spanish [18], Mandarin [20], Hebrew and Arabic [21], and Yoruba versions [19] of the LEAQ. This method allowed for errors to be identified, and it improved the readability of the questionnaire by eventually developing two questionnaires, one for males and one for females. The translation process was a result of multidisciplinary teamwork between translators and healthcare professionals.

To evaluate the reliability and validity of the Maltese version of the LEAQ, scale and item analysis were carried out. Results of the scale analysis showed that the Maltese version of the LEAQ showed satisfactory age-dependency since a high correlation (0.90) between the age of the children and the total score was observed. The older the child, the higher the child's expected score was. This provides evidence for the validity of the Maltese version of the LittlEARS® questionnaire. When compared to other language versions such as Mandarin [20], Yoruba [19], and Persian [22], the Maltese version shows one of the highest correlation coefficients and has a very similar coefficient value to the original German version (0.91) [17].

Scale analysis also showed high internal consistency. Responses from subjects were greatly consistent across the questionnaire items. The Cronbach's alpha coefficient value

is 0.92 for the Maltese LEAQ and, similarly, 0.96 for the original German version [17], 2009). This is also evident across other LEAQ versions, which all show internal consistency values higher than 0.90. This suggests that in Maltese and interlingually, LittlEARS® can reliably differentiate the degree of auditory development in children. This is important since hearing development predictors should not differ among children speaking different languages.

The split-half reliability coefficient for the Maltese version of LittlEARS® was 0.92 which indicates a high measuring accuracy of the questionnaire. The Maltese LEAQ shows slightly higher scores than the original German version (0.88) [17] and other languages such as Yoruba (0.70) [19] and Persian (0.73) [22]. The Mandarin version also shows excellent reliability, with a 0.91 reliability coefficient [20].

Lastly, scale analysis showed high predictive accuracy, with a value of 0.92. This is very similar to the German (0.93) [17], Turkish (0.91) [23] and Mandarin versions (0.88) [20]. The Persian version [22] has a higher predictive accuracy (0.96), whilst the Yoruba [19] has a low but satisfactory value (0.58). The high predictive accuracy of the Maltese version suggests that the dependent variable, in this case the total score on the LEAQ, can be accurately predicted based on the independent variable, which is the age of the child. This has important implications for the use of the LEAQ in a clinical setting, whereby the expected scores of children at a certain age are compared to the actual scores of the child, and clinical decisions are made accordingly.

Regarding item analysis, the results of this study showed that the index of difficulty is slightly higher on the Maltese version (0.65) when compared to the original German version (0.63) [17]. These findings are similar across the Mandarin (0.68) [20] and the Spanish version (0.64) [18]. A high difficulty index for an item indicates that it reflects the earliest-developed behaviours and, thus, less complex auditory skills. The items on the LEAQ become more difficult as the questionnaire progresses, with more complex items at the end of the questionnaire. This occurs as the auditory skill necessary becomes more advanced. As mentioned in the results section, items with a high index of difficulty were kept in the questionnaire in order to avoid zero-point scores, as this puts less pressure on the parents who are filling out the questionnaire. The low discriminatory power of the initial questions is similar across language versions such as German [17], Mandarin [20], Spanish [18], and Yoruba [19].

The correlation of item score to the child's age was investigated to check the items' suitability for measuring the age dependency of behaviour. About one third of the LEAQ items showed a strong positive correlation with age (r ≥ 0.7), confirming that parents report few auditory abilities at a young age and an increasing number of auditory skills as they get older. As also reported in Offei's study [24]), this study confirms that the score remains fairly level after the age of 24 months in hearing children (N = 131). The results of this study show that the Maltese version of the LEAQ reaches maximum values between the age of 24 and 36 months. Over 92% of children above 24 months of age score between 32 and 35, confirming that ceiling effects remain stable. This finding shows that the Maltese version of LEAQ might be an efficient tool for the screening of children above 24 months of age, along with other tests that are used routinely in an audiology clinic or school setting for the assessment of young children. However, further research in this area is warranted in the local population.

In summary, by comparing the Maltese version with the original German version and other language versions, this research shows that the psychometric properties of the Maltese version of the LEAQ are excellent and indicate high reliability and validity as a tool to measure auditory behaviour in children less than 2 years of age.

### 4.2. Normative Values of the Maltese LEAQ

Normative data was generated from the total scores of the participants and their ages and visually displayed as a norm curve. The average score for a particular child's age provided the expected value, whilst the lower limit of the 95% confidence interval provided

the minimum values. The regression equation for the Maltese sample is y= $-0.03x^2 + 2.02x + 5.07$, whilst the regression equation of the German data is y = $-0.038x^2 + 2.22x + 2.07$. The coefficient of x and $x^2$ are very similar, explaining why the two curves are almost parallel. The constant terms (5.07 and 2.07) differ by 3, implying that the Maltese children are scoring 3 points higher, on average, than those from the German data. The percentage of explained variance was 86% for the original German version and 82% for the Maltese LEAQ. This means that age is slightly more predictive of the total score for the German version than for the Maltese one. These slight differences in statistics could be due to the sample composition and mode of administration used in this research study. Overall, the original norm curve developed in German is very similar to the norm curve developed in this research study in Maltese.

Table 3 also shows the regression equation of several other language versions. One can note that the coefficients are very similar across most of the languages, especially among the German, Maltese and Mandarin versions, as well as the multilingual study by Coninx [17]. Similarly, the constants of the Maltese norm curve differ very slightly (5.07 and 3.47), confirming how close the Maltese norm curve is to the 15 other languages validated by Coninx et al. [17] when compared to other language versions.

A more noticeable difference is found between variance coefficients across other language versions. Whilst similar variance is seen in languages such as Polish (83%), Spanish-USA (81%), Spanish (79%), and Persian (80%), larger differences from the original German version are apparent in other languages such as Mandarin (73%), Turkish (74%), Yoruba (75%), Hebrew (85.7%), and Arabic (72.7%). Overall, one can conclude that the Maltese norm curve is very similar to that of the multilingual study validated by Coninx et al. [17], German and Mandarin.

In conclusion, the results indicated that the Maltese version of the LEAQ is a valid and reliable outcome tool in the Maltese-speaking population. This study supports the use of the LEAQ for making informed decisions in a clinical or educational setting, as it makes valid inferences about the child's auditory development in the first 2 years of life. This is due to the excellent psychometric properties of the Maltese version, including the high correlation between age and score. Responses that fall below the minimum values would alert the professionals involved with the child, such as paediatricians and speech and language therapists, to refer them for an audiological evaluation. Most importantly, the LEAQ has the potential to fill in an important gap in the screening and diagnosis of hard-of-hearing children between birth and school age [5]. In Malta, developmental assessments for babies are carried out at 3 routine visits at 6 weeks, 8 months, and 18 months, respectively. These visits are done at the Well Baby Clinics, which are available at the community level, making it a very accessible service. At these visits, clinical examinations are carried out to evaluate whether the child has reached certain developmental milestones. Including a hearing screen at one of these stages would potentially aid in identifying children with hearing loss who were not identified at birth. The Maltese version of the LEAQ would ideally be carried out at the 18 months visit. At 8 months of age, a limited number of 'yes' responses may be scored on the LEAQ, which would make it less effective as a screening instrument. The Well Baby service is also available for babies born in a private hospital, thus enabling a larger number of children to be screened.

Post-UNHS screening is indeed becoming an essential step in the identification and management of hearing loss, and this is only possible through reliable tools such as the LEAQ [5]. This is crucial in the identification of hearing loss that is late-onset, acquired, or not detected in newborns. Early identification supports speech and language development and literacy skills [25,26].

Future research is also recommended in relation to the use of the Maltese version of the LEAQ in monitoring children's progress with hearing aids and cochlear implants and in planning rehabilitation programmes.

**Author Contributions:** Conceptualization, F.C.; methodology, F.C. and P.M.; software, F.C.; validation, P.M.; formal analysis, F.C. and P.M.; investigation, F.C.; resources, F.C. and P.M.; data curation,

F.C., P.M. and K.S.; writing—original draft preparation, P.M.; writing—review and editing, K.S. and F.C.; visualization, P.M., F.C. and K.S.; supervision, F.C. and K.S.; project administration, F.C.; funding acquisition, N/A. All authors have read and agreed to the published version of the manuscript.

**Funding:** This research received no external funding.

**Institutional Review Board Statement:** The study was conducted in accordance with the Declaration of Helsinki and approved by the University of Malta research ethics committee (UREC) (Ref. No. UREC-DP1801016EXT, 2 April 2018).

**Informed Consent Statement:** Informed consent was obtained from all subjects involved in the study.

**Data Availability Statement:** The data presented in this study are available on request from the corresponding author.

**Conflicts of Interest:** The authors declare no conflict of interest.

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
