# Peer review of "Validation of the LittlEARS® Questionnaire in Hearing Maltese-Speaking Children"

_audiolres, doi:10.3390/audiolres12020022_

Round 1

Reviewer 1 Report

Thank you for the invitation to review "Validation of the LittlEARS Questionnaire in hearing Maltese-speaking children." This paper adequately validates a caregiver questionnaire for auditory development in Maltese. The development of valid and reliable tools for use in languages other than English is greatly needed and you have answered that call. I have only minor recommendations to improve your work.

Introduction:

Pg 1 line 31: Many studies and EHDI programs use the LEAQ to monitor auditory development in children already diagnosed with hearing loss. Although being delayed on the LEAQ could mean hearing loss, it is clearly not a hearing screener. Is this how you meant to frame the use of the tool?

Pg 1 line 42: Given that most families in Malta are bilingual, should they be assessed bilingually?

Pg 2 line 44: Caregivers have been known to be bias when reporting on their child's development. Do you think that influences LEAQ scores?

Methods:

Pg 2 line 62: Did the socioeconomic status of the participants reflect the wider Maltese population? SES has been shown to influence communication development. What languages were used by the participants? Did their language use reflect the population numbers you reported?

Pg 2 line 64: Was the LEAQ given on paper or on a computer? Were other versions of the LEAQ presented by clinicians/researchers or are parents always asked to complete it without support?

Pg 2 line 71: It becomes clear why children older than 24 months were included in the study when you report the results but it is unclear why you've included them here since the LEAQ only goes up to 24 months. I would add a clarification sentence or two here.

Pg 3 line 83: What level of training did the professional translators have? Did they have familiarity with communication disorders and development?

Pg 3 line 92: This is interesting. Are there other LEAQ translations that require two versions for gender? Readers will likely not be familiar with Maltese so a little more explanation might be helpful.

Results:

Pg 4 line 140: Given that you translated the LEAQ from English, why choose to compare results to the German over the English version. In fact, I noticed that the statistics for the validation of the English version don't seem to be in the article. Is there a reason?

Page 4 line 144: Please refer to Table 2 in the body of the manuscript.

Page 4 line 178: "compares" should be "compare"

Page 7 line 201: Is there a significant difference between the three curves (Maltese, German, 15 languages)?

Author Response

Reviewer 1

Introduction:

Pg 1 line 31: Many studies and EHDI programs use the LEAQ to monitor auditory development in children already diagnosed with hearing loss. Although being delayed on the LEAQ could mean hearing loss, it is clearly not a hearing screener. Is this how you meant to frame the use of the tool?

LEAQ has already been used as a hearing screening tool in Germany by my tutors Prof Schaefer and Prof Coninx in 2019. Kindly see reference below.

Karolin Schaefer, Frans Coninx & Thomas Fischbach (2019) LittlEARS auditory questionnaire as an infant hearing screening in Germany after the  newborn hearing screening, International Journal of Audiology, 58:8, 468-475, DOI: 10.1080/14992027.2019.1597287

Pg 1 line 42: Given that most families in Malta are bilingual, should they be assessed bilingually?

Whilst most families are bilingual, the majority are predominantly Maltese speaking. Non Maltese speaking parents are able to use the English version of LittlEARS.

Pg 2 line 44: Caregivers have been known to be bias when reporting on their child's development. Do you think that influences LEAQ scores?

It is indeed a limitation of all parental questionnaires, and it is taken into account by the authors.

Methods:

Pg 2 line 62: Did the socioeconomic status of the participants reflect the wider Maltese population? SES has been shown to influence communication development. What languages were used by the participants? Did their language use reflect the population numbers you reported?

The parental questionnaires were collected from a number of childcare centres in different areas of Malta, which reflect different SES. Participants who understood written Maltese were eligible to take part, even though they may have had a different first language such as English.

Should I make a note of this in the text?

Pg 2 line 64: Was the LEAQ given on paper or on a computer? Were other versions of the LEAQ presented by clinicians/researchers or are parents always asked to complete it without support?

The LEAQ was presented on a paper always, and always without support.

The aim was to confirm ceiling effects in the third year of life. Edited text accordingly.

Pg 3 line 83: What level of training did the professional translators have? Did they have familiarity with communication disorders and development?

Yes, they work as professional translators and in an educational setting

Pg 3 line 92: This is interesting. Are there other LEAQ translations that require two versions for gender? Readers will likely not be familiar with Maltese so a little more explanation might be helpful.

There are no other languages with two versions.

In Maltese both nouns and verbs reflect gender, which may complicate translation. Item 2 in LittlEARS, ‘Does your child hear somebody speak?’ would be translated as ‘Ibnek/bintek jisma’/tisma’ lil xi ħadd jitkellem?’. Hence, two versions were necessary to simplify the text for the readers.

A note in the text was added.

Results:

Pg 4 line 140: Given that you translated the LEAQ from English, why choose to compare results to the German over the English version. In fact, I noticed that the statistics for the validation of the English version don't seem to be in the article. Is there a reason?

Since the original norms were in German, it was compared to the original one. This was the way forward in several LittlEARS validation studies.

The English version is the standard version used for translating LittlEARS.

Page 4 line 144: Please refer to Table 2 in the body of the manuscript.

I could not find any notes in the manuscript. Kindly clarify.

Page 4 line 178: "compares" should be "compare"

Noted and fixed.

Page 7 line 201: Is there a significant difference between the three curves (Maltese, German, 15 languages)?

The significant difference between the three curves was not tested.

Reviewer 2 Report

I appreciate the opportunity to review the manuscript “Validation of the LittlEARS® Questionnaire in hearing Maltese-speaking children”. The topic is appropriate for the Audiology Research. Although the rationale for the study is certainly warranted, the paper currently have a few flows that need to be addressed.

 First concern

The aim of the study is: “The present cross-sectional study aims to translate and validate the LEAQ for hearing Maltese infants and to define Maltese critical score values (expected and minimum). It also aims to provide a tool that may be used as a second hearing screening following the newborn hearing screening”.

The first aim is obviously achieved, with the second one there is a serious concern. There is no data (eg. sensitivity and specificity of the test) no strict criteria of passing or failing the test, that should be provided to finally claim that a tool can serve as a screening test. I suggest to remove the second sentence from the aim. In the discussion authors claim: “This finding shows that the Maltese version of LEAQ might be an efficient tool for the screening of children above 24 months of age, along with other tests that are used routinely in an audiology clinic or school setting for the assessment of young children. However, further research in this area is warranted in the local population.” (page 9, lines: 298-302)

The statement that, the tool might be an efficient tool  and that further research is needed is appropriate in the context of current study.  Nevertheless information that it is suitable for children above 24 months of age  generates the second concern.

Second concern

The LittlEARS is intended to assess auditory development in children up to 24 months of age. The motivation for the assessment of children older than 2 years with the questionnaire is not clear. In the method section it was mentioned that the authors would like to use it for screening purpose in older children, in conclusion (page 10, lines 336-356) the authors postulate assessment at 18 months (not consistent). Please explain in the method section why children above the age of 2 were tested, why there is a Kruskal–Wallis was performed (for children up to 24 months there is a strong correlation with age, so significant differences are obvious, for older children there is a ceiling effect because the questionnaire is not measuring skills that are developing later). I suggest to stronger motivate the analysis in children over 24 months of age or to remove those from the manuscript. Also note that use of the questionnaire in a new population, that is not intended for, requires separate validation study. Moreover, Kruskal–Wallis test is non parametric test it is testing whether samples originate from the same distribution, no the differences between means. It was used here probably because the data were not normally distributed (also not described), if so, median is better measure of central tendency than mean, therefore the median should be reported.

Third concern

Selection of participants. I agree that random selection is the best option from statistical point of view. In case of real randomization of participants much more data should be provided: exact number of children in selected age range, how did you get data about this population, how did you calculate the number of participants that is exactly 398, how did you get in contact with the patients, how many patients did you contact, how many refused to participate, how did you manage to distribute questionnaires together with the person who performed OAE to hospitals and childcare centers all over country. If only selected hospitals and childcare centers were involved, please write how many, and the period of data collection. – Please describe the method of patients’ selection.

Additional comments

Keywords: remove “3” form keywords

Introduction: please provide link between first and second paragraph.

Page 9, lines 303-306: “In summary, by comparing the Maltese version with the original German version and other language versions, this research shows that the psychometric properties of the Maltese version of the LEAQ are excellent and indicate high reliability and validity as a screening tool of auditory behaviours in children less than 2 years of age”. The information on questionnaire validity as a screening too is not valid (see First concern). Pleas rewrite the sentence.

Table 1: Clip in the table is too short suggesting that only results of children aged 6-23 were analyzed, please correct.

Page 3, lines: 101-106 – this paragraph is not clear, please rewrite it.

Page 4, line 119. Please describe in more details what measures were used to evaluate psychometric properties, scale and item analysis.

Page 4, line 152,153: “This shows that the items of the questionnaire are presented in order of difficulty”  - This is not exactly the case. The difficulty index is almost the same for all items accept for 1,2,3,4,5 and 33. Please rewrite the sentence and explain why the difficulty index is not working similarly to original data.

Page 10: line 340, 341: “Responses that fall below the minimum and expected values would alert the professionals involved with the child…” Falling below expected values is not the reason for rising alert. The authors previously stated that results above minimum value indicate high probability of age appropriate auditory development (page 3, lines 15,116) – please correct the sentence.  

Author Response

Reviewer 2

First concern

The aim of the study is: “The present cross-sectional study aims to translate and validate the LEAQ for hearing Maltese infants and to define Maltese critical score values (expected and minimum). It also aims to provide a tool that may be used as a second hearing screening following the newborn hearing screening”.

The first aim is obviously achieved, with the second one there is a serious concern. There is no data (eg. sensitivity and specificity of the test) no strict criteria of passing or failing the test, that should be provided to finally claim that a tool can serve as a screening test. I suggest to remove the second sentence from the aim. In the discussion authors claim: “This finding shows that the Maltese version of LEAQ might be an efficient tool for the screening of children above 24 months of age, along with other tests that are used routinely in an audiology clinic or school setting for the assessment of young children. However, further research in this area is warranted in the local population.” (page 9, lines: 298-302)

The statement that, the tool might be an efficient tool  and that further research is needed is appropriate in the context of current study.  Nevertheless information that it is suitable for children above 24 months of age  generates the second concern.

LEAQ has already been used as a hearing screening tool in Germany by my tutors Prof Schaefer and Prof Coninx in 2019. That is the main reason behind my statement. Is there anything I may add to make it more apparent?

Karolin Schaefer, Frans Coninx & Thomas Fischbach (2019) LittlEARS auditory questionnaire as an infant hearing screening in Germany after the  newborn hearing screening, International Journal of Audiology, 58:8, 468-475, DOI: 10.1080/14992027.2019.1597287

Second concern

The LittlEARS is intended to assess auditory development in children up to 24 months of age. The motivation for the assessment of children older than 2 years with the questionnaire is not clear. In the method section it was mentioned that the authors would like to use it for screening purpose in older children, in conclusion (page 10, lines 336-356) the authors postulate assessment at 18 months (not consistent). Please explain in the method section why children above the age of 2 were tested, why there is a Kruskal–Wallis was performed (for children up to 24 months there is a strong correlation with age, so significant differences are obvious, for older children there is a ceiling effect because the questionnaire is not measuring skills that are developing later). I suggest to stronger motivate the analysis in children over 24 months of age or to remove those from the manuscript. Also note that use of the questionnaire in a new population, that is not intended for, requires separate validation study. Moreover, Kruskal–Wallis test is non parametric test it is testing whether samples originate from the same distribution, no the differences between means. It was used here probably because the data were not normally distributed (also not described), if so, median is better measure of central tendency than mean, therefore the median should be reported.

The aim was to confirm ceiling effects in the third year of life. Edited text accordingly.

The sample had a non normal distribution, and therefore a Kruskal Wallis test was used. Should I also include the median? Kindly clarify.

Edited in text

Third concern

Selection of participants. I agree that random selection is the best option from statistical point of view. In case of real randomization of participants much more data should be provided: exact number of children in selected age range, how did you get data about this population, how did you calculate the number of participants that is exactly 398, how did you get in contact with the patients, how many patients did you contact, how many refused to participate, how did you manage to distribute questionnaires together with the person who performed OAE to hospitals and childcare centers all over country. If only selected hospitals and childcare centers were involved, please write how many, and the period of data collection. – Please describe the method of patients’ selection.

Table 1 shows the exact number of children in selected age range. Is there anything else I should add?

I got birth data from our national statistics office and calculated approximately how may participants are needed to get a good enough sample. https://www.surveysystem.com/sscalc.htm. 95% confidence and 5% CI with a population of 12,900 equals a sample size of 373. I managed to get some more, 398. Not sure if it was clear in the text.

 I contacted several childcare centres myself via email and those who agreed were included in the study. I carried out the research by myself since it was my Phd study.

The number of childcare centers involved and the period of data collection was added in the text.

Additional comments

Keywords: remove “3” form keywords

Noted. removed

Introduction: please provide link between first and second paragraph.

Adjusted text accordingly.

Page 9, lines 303-306: “In summary, by comparing the Maltese version with the original German version and other language versions, this research shows that the psychometric properties of the Maltese version of the LEAQ are excellent and indicate high reliability and validity as a screening tool of auditory behaviours in children less than 2 years of age”. The information on questionnaire validity as a screening too is not valid (see First concern). Pleas rewrite the sentence.

Edited in text

Table 1: Clip in the table is too short suggesting that only results of children aged 6-23 were analyzed, please correct.

adjusted

Page 3, lines: 101-106 – this paragraph is not clear, please rewrite it.

Rewritten

Page 4, line 119. Please describe in more details what measures were used to evaluate psychometric properties, scale and item analysis.

Edited

Page 4, line 152,153: “This shows that the items of the questionnaire are presented in order of difficulty”  - This is not exactly the case. The difficulty index is almost the same for all items accept for 1,2,3,4,5 and 33. Please rewrite the sentence and explain why the difficulty index is not working similarly to original data.

I have gone back to the raw data and re-evaluated them. It appears that I have either copied in an erroneous measurement or calculated it incorrectly. I have attached the raw data as proof. The raw data confirms that the items in the questionnaire are presented in order of difficulty.

Page 10: line 340, 341: “Responses that fall below the minimum and expected values would alert the professionals involved with the child…” Falling below expected values is not the reason for rising alert. The authors previously stated that results above minimum value indicate high probability of age appropriate auditory development (page 3, lines 15,116) – please correct the sentence.  

Edited
